# Satisfactory In Vitro Activity of Ceftolozane–Tazobactam against Carbapenem-Resistant *Pseudomonas aeruginosa* But Not against *Klebsiella pneumoniae* Isolates

**DOI:** 10.3390/medicina59030518

**Published:** 2023-03-07

**Authors:** Alicja Sękowska, Marta Grabowska, Tomasz Bogiel

**Affiliations:** 1Microbiology Department, Ludwik Rydygier Collegium Medicum in Bydgoszcz, Nicolaus Copernicus University, Torun, 85-094 Bydgoszcz, Poland; 2Clinical Microbiology Department, Dr Antoni Jurasz University Hospital No 1 in Bydgoszcz, 85-094 Bydgoszcz, Poland; 3Dr Jan Biziel University Hospital No 2 in Bydgoszcz, 85-168 Bydgoszcz, Poland

**Keywords:** carbapenemases, carbapenems, ceftolozane, ceftolozane–tazobactam, Gram-negative rods, imipenem, *Klebsiella pneumoniae*, meropenem, *Pseudomonas aeruginosa*, tazobactam

## Abstract

*Background*: Gram-negative rods are one of the most commonly isolated bacteria within human infections. These microorganisms are typically opportunistic pathogens that pose a serious threat to public health due to the possibility of transmission in the human population. Resistance to carbapenems is one of the most important antimicrobial resistance mechanisms amongst them. The aim of this study was to evaluate ceftolozane–tazobactam in vitro activity against carbapenem-resistant *Pseudomonas aeruginosa* and *Klebsiella pneumoniae* clinical strains. Information on the antimicrobial activity of this antimicrobial against Gram-negative rods was also supplemented with a brief review of the relevant literature. *Methods*: The research involved 316 strains of Gram-negative rods: *P. aeruginosa*—206 and *K. pneumoniae*—110. *Results*: Of the tested strains, 86.0% *P. aeruginosa* and 30.0% *K. pneumoniae* remained susceptible to ceftolozane–tazobactam. *Conclusions*: Therefore, ceftolozane–tazobactam might be a good option in the treatment of infections caused by carbapenem-resistant *P. aeruginosa* strains, including those in ICU patients. Meanwhile, due to dissemination of ESBLs among *K. pneumoniae* strains, infections with this etiology should not be treated with the ceftolozane–tazobactam combination.

## 1. Introduction

*Pseudomonas aeruginosa* and *Klebsiella pneumoniae* isolates are currently one of the most frequent causes of infections of immunocompromised patients [1]. These rods are also among the most frequently isolated bacteria from infections caused by Gram-negative bacteria, which include pathogens often causing nosocomial and opportunistic infections [2,3,4]. *P. aeruginosa* shows intrinsic lack of susceptibility to a number of antimicrobials; both species easily acquire new resistance mechanisms [5]. These include resistance to carbapenems, often administered as the last-choice drugs [2,6]. The increasing emergence of multi-drug-resistant *P. aeruginosa* and *K. pneumoniae* strains limits the treatment options, which increases the morbidity and mortality of patients [7]. This situation enforces the search and more frequent application of new drugs such as ceftolozane–tazobactam, new drug combinations, or new applications of commonly known antimicrobials [3,7,8,9,10,11,12].

Ceftolozane–tazobactam (C/T) is a relatively new antimicrobial agent approved for the treatment of complicated urinary tract infections, including acute pyelonephritis [8]. It is also applied in the treatment of complicated intra-abdominal infections (in combination with metronidazole) and nosocomial pneumonia (including ventilator-associated pneumonia) [8]. It is characterized by an activity against Enterobacterales and *P. aeruginosa* rods [4,13,14,15,16].

Ceftolozane–tazobactam consists of a new cephalosporin (ceftolozane) and class A β-lactamase inhibitor (tazobactam). This antimicrobial shows high in vitro activity against multi-drug-resistant *P. aeruginosa* strains and *Enterobacteriaceae* family representatives producing extended-spectrum β-lactamases (ESBL) [13,15,17,18]. Due to the presence of pyrazole in the ceftolozane side chain, the drug shows activity against pathogens producing AmpC-type enzymes [19,20]. Moreover, it is characterized by high affinity for penicillin-binding proteins (PBP). Ceftolozane–tazobactam has been shown to be effective in the treatment of complicated urinary tract infections caused by *P. aeruginosa*, *Escherichia coli*, *K. pneumonia,* and *Proteus mirabilis* and (in combination with metronidazole) in the treatment of complicated intra-abdominal infections caused by *P. aeruginosa*, *E. coli*, *K. pneumoniae*, *P. mirabilis*, *Enterobacter cloacae*, *Klebsiella oxytoca*, *Streptococcus anginosus*, *Streptococcus constellatus*, *Streptococcus salivarius,* and *Bacteroides fragilis* [21]. The breakpoints of ceftolozane–tazobactam minimum inhibitory concentration (MIC, mg/L) according to EUCAST are currently set at the level of: ≤4 (susceptible) and >4 (resistant) for *P. aeruginosa* ≤ 2 (susceptible) and >2 (resistant) for *K. pneumoniae* [22].

The aim of the study was to evaluate in vitro activity of the combination of ceftolozane–tazobactam against clinical strains *P. aeruginosa* and *K. pneumoniae* resistant to at least one of the carbapenems and to briefly review the relevant literature concerning the study’s topic.

## 2. Results

### Results of Ceftolozane–Tazobactam Susceptibility Tests

A number of isolates were isolated from the same patients (up to 7). Therefore the remaining strains were excluded from further analysis, and the final numbers of strains included in the percentage calculation of susceptibility to ceftolozane–tazobactam were as follows: 150 for *P. aeruginosa* and 100 for *K. pneumoniae*. Among 150 *P. aeruginosa* strains, 129 (86.0%) were susceptible to ceftolozane–tazobactam, while 21 (14.0 %) were resistant to this antimicrobial. The corresponding values for 100 *K. pneumoniae* were 30 (30.0%) and 70 (70.0%), respectively. C/T MIC distribution for the particular groups of strains is shown in Figure 1. The detailed susceptibility profiles of the examined *P. aeruginosa* strains are presented in Appendix A, whereas *K. pneumoniae* is presented in Appendix A. The strains recognized as repeating isolates from the same patient were excluded from calculations.

## 3. Discussion

These days, in which the resistance rate to antimicrobials is increasing, the search for new drugs and monitoring susceptibility to antibiotics are of utmost importance. There are available articles concerning reactivity to ceftolozane–tazobactam, but only a few refer to isolates derived from Central and Eastern Europe. The only report considering the isolates from Poland was conducted by Saran et al. [23] on a relatively small group of the strains derived from hematological patients. At the same time, C/T is one of the few β-lactam representatives accepted for the treatment of infections caused by multidrug resistant Enterobacterales and *P. aeruginosa*.

C/T activity differs significantly between *P. aeruginosa* strains, mostly depending on the resistance profile of the investigated strains. One of the first studies on C/T activity conducted by Shortridge et al. [13] on a large group of 3851 strains from US medical centers (2012–2015) revealed that the strain susceptibility (MIC ≤ 4 mg/L) was noted amongst 97.0% of the overall number of the strains (with only 51 isolates resistant to this antimicrobial). The corresponding values for the groups of meropenem-non-susceptible isolates simultaneously non-susceptible to the β-lactams piperacillin/tazobactam; ceftazidime, cefepime, and meropenem; MDR isolates; and XDR strains reached 87.6%, 68.0%, 84.0%, and 76.9%, respectively. The obtained results indicate that C/T is the most active beta-lactam tested.

In the following research, Shortridge et al. [24] obtained very similar results to the analyses performed for the isolates derived from Australia and New Zealand (2016–2018). Among 435 *P. aeruginosa* strains, 97.5% were susceptible to C/T. High percentages of strains were also found susceptible to meropenem (89.9%) and piperacillin/tazobactam (86.4%). Interestingly, the only antimicrobial from the aforementioned group of strains that exceeded C/T activity was colistin in the study in the USA (over 98.1%) and colistin in research in Australia and New Zealand (99.3%). In turn, a relatively low percentage of susceptibility of *P. aeruginosa* strains to C/T was reported by García-Betancur et al. [25]. Assessing the susceptibility of 508 strains isolated from patients in five Latin American countries, it reached 68.1% and was the highest among beta-lactam antibiotics.

Having conducted the research, Sutherland et al. [26] analyzed *P. aeruginosa* strains derived exclusively from samples from the respiratory tract. Among 144 isolates, 94.0% were susceptible to C/T, and 14.0% were defined as MDR. A high percentage of strains was confirmed susceptible to tobramycin (91.0%) and ceftazidime (78.0%) too. Similar results were obtained by Tantisiriwat et al. [27] for the strains isolated from different types of clinical samples. Among 100 isolates, 94.0% were susceptible to C/T, and 18.0% were defined as MDR. Among MDR strains, 55.6% were resistant to C/T, while 16.7% of them synthesized carbapenemases. In our study, the strains were cultured from different clinical specimens, including blood, respiratory tract, urine samples, and wound swabs. The C/T susceptibility rate was lower, at exactly 86%, while a similar number of isolates was analyzed with 56% being MDR. High activity was estimated for two non-β-lactam antibiotics: colistin, at above 84%, and amikacin, at almost 63%. Similar results to the ones obtained in this study were received by Gonzalez et al. [28], who conducted a study of 45 *P. aeruginosa* strains. The authors noted that 87.0% of strains were susceptible to C/T. The specimens were of different origins: the respiratory tract, blood, urine, and wounds. Grupper et al. [29] conducted interesting research rating susceptibility to C/T of *P. aeruginosa* strains with respect to their origin. The strains cultured from blood, wound swabs, and the respiratory tract presented C/T susceptibility at the levels of 89.0%, 88.0%, and 92%, respectively. In turn, Klinker et al. [30] assessed the sensitivity of strains isolated from blood and respiratory tract material and recorded 88.5% of strains as sensitive to C/T. The observed percentage was significantly higher than for other β-lactams—piperacillin/tazobactam (57.7%), cefepime (55.8%), and meropenem (67.9%).

In turn, Sader et al. [31] analyzed a large group of *P. aeruginosa* strains (7503) isolated between 2012 and 2018 from different medical centers from Western and Eastern Europe. The authors detected 92.8% to 96.2% (average of 94.1% strains from Western Europe) and 71.4% to 87.4% (average of 80.9% from Eastern Europe) of the strains as susceptible to C/T. Interestingly, the researchers observed a huge difference in C/T MIC_90_ values between the isolates cultured in Eastern and Western Europe, at >32 mg/L and 2 mg/L, respectively. Meanwhile, the C/T MIC_50_ values were similar at 1 mg/L and 0.5 mg/L, respectively. Sader et al. [31] noted that general susceptibility percentage for all the analyzed antimicrobials, except for colistin, were higher in Eastern Europe than in Western Europe. Additionally, high percentages of susceptibility to colistin (above 99%) and to tobramycin (70.9–88.6%) were detected. This study also included data from Poland pointing out that among 445 *P. aeruginosa* strains, 365 (82.0%) were susceptible to C/T. Data regarding susceptibility to C/T from Eastern Europe varied from 41.2% in Belarus (68 strains) to 91.3% in Slovenia (23 strains), while the corresponding value in Israel reached 98.2%.

Pfaller et al.’s study [32] revealed C/T as the most potent β-lactam agent, tested on 537 *P. aeruginosa* isolates. Over 86% of the isolates were susceptible to C/T (MIC ≤4 mg/L). Based on the MIC_50_ (0.5 mg/L) value, C/T was 2-fold more active than meropenem, 8-fold more active than cefepime and ceftazidime and 16-fold more active than piperacillin/tazobactam. The results are fairly consistent with the results of our study.

Livermore et al. [17] detected 99.8% of *P. aeruginosa* isolates (1.097 out of 1.099) susceptible to C/T, whereas susceptibility rates to other antimicrobials were as follows: 97.9% for gentamicin, 97.6% for ceftazidime, 95.5% for piperacillin/tazobactam, 91.7% for meropenem, 92.5% for imipenem, and 90.4% for ciprofloxacin. One of the two ceftolozane–tazobactam-resistant *P. aeruginosa*, with a MIC value of 32 mg/L, synthesized a VIM-type carbapenemase, while the resistance mechanism remained uncertain in the other one, with a MIC value of 8 mg/L. C/T was the most active β-lactam agent with MIC C/T four-fold lower than the one of ceftazidime. In the cited study, 99.7% of isolates with moderately active efflux and 94.7% with strongly activated efflux were susceptible to C/T. C/T was active against 96.6% of the isolates with de-repressed AmpC, whereas ceftazidime was active against only 20.8% of them, rising up to 94.5% for ceftazidime/avibactam combination. In contrast to this good activity against isolates with elevated efflux activity, for de-repressed AmpC or loss of OprD proteins, resistance to C/T was very high (reaching 96.8–100%) also among *P. aeruginosa* with metallo-beta-lactamases (MBLs) or Vietnamese extended-spectrum β-lactamases (VEB-type ESBLs). Isolates with these enzymes were resistant to penicillins and cephalosporins. C/T was active against 16/19 of Guiana extended-spectrum β-lactamase (GES) carbapenemase-positive isolates, compared with 74% sensitivity for ceftazidime; however, 14 of these 19 isolates were derived from a single outbreak, and the results may not be representative. Aside from MBL and ESBL producers, the only *P. aeruginosa* groups where C/T resistance was frequent were the unassigned categories with ceftazidime MICs of 16–128 and 256 mg/L. C/T MICs for these isolates mostly were 8–16 mg/L, thus slightly lower than for MBL and ESBL producers. The differences in the presented work and quoted could be associated with different numbers and times, where strains were isolated, various clinical specimens, and also with different number of MDR isolates. Sid Ahmed et al. [33] also evaluated the C/T sensitivity of *P. aeruginosa* strains producing carbapenemases belonging to different classes. The highest percentages of strains sensitive to this drug were recorded for bacteria producing class D carbapenemases (100%), followed by class C (97.1%) and class A (22.9%). None of the tested strains of *P. aeruginosa* producing class B carbapenemases were sensitive to C/T.

An interesting aspect has been shown by Sader et al. [34] rating susceptibility to C/T strains isolated from patients of intensive care unit (ICU) and non-ICU. In a general population of the analyzed strains, the authors did not observe significant differences, the susceptibility levels for ICU-derived strains reached 96.1%, while for non-ICU-derived strains, the value was 98.2%. Among MDR and XDR isolates, the corresponding values reached 80.3% and 89.8% and 74.5% and 85.0%, respectively. In this study, the susceptibility rates between isolates from ICU patients and non-ICU patients were above 91% and 82%, respectively. Caffrey et al. [11] noted significantly lower mortality rate for patients infected with *P. aeruginosa* receiving C/T compared to the group of patients treated with aminoglycosides or polymyxins (15.8% vs 27.7%). At the same time, in the group of patients infected with MDR *P. aeruginosa* strains, mortality was 61% lower than in patients treated with aminoglycosides or polymyxins. In turn, Gallagher et al. [35] reported higher a cure rate and lower nephrotoxicity rate in patients treated with C/T than in patients treated with aminoglycosides or polymyxins.

In one of the initial studies of C/T activity, conducted by Pfaller et al. [32], C/T MIC values ranged from ≤0.015 to >32 mg/L. Over 84% of the tested *Enterobacteriaceae* isolates (1878 altogether) were inhibited at a MIC value of ≤2 mg/L, while 85.9% were inhibited at ≤4 mg/L. C/T showed good activity against ESBL-positive non-CRE phenotype *Enterobacteriaceae* strains (74.7% remained sensitive according CLSI and 66.9% according to EUCAST). However, C/T did not present satisfactory activity against carbapenem-resistant isolates (only 1.6% remained sensitive). *Enterobacteriaceae* isolate susceptibility rates to other β-lactam agents ranged from 63.4% for cefepime to 61.8% for ceftazidime, 75.7% for piperacillin/tazobactam, and 93.6% for meropenem (using EUCAST breakpoints). Only meropenem, amikacin, colistin, and C/T retained clinical activity against 495 ESBL-positive non-CRE *Enterobacteriaceae* isolates. In the cited study, C/T also showed potent activity against isolates of *K. pneumoniae* and retained activity against many ESBL-positive non-CRE phenotype isolates (60.4/46.0% remained susceptible).

In the study of Livermore et al. [17], the researchers provided a random and geographically diverse collection of bloodstream-derived isolates. The data pooled from 2011 to 2015 revealed that, using EUCAST breakpoints (4 mg/L), susceptibility rates to C/T reached 97.6% for *Klebsiella* spp., exceeding those for all other β-lactams tested, except for carbapenems. The susceptibility rates for ESBL-producing *Klebsiella* spp. were relatively low (84–85%). Around half of the C/T-resistance among ESBL-producing *Klebsiella* representatives were at low level, with MICs of 2 mg/L, but other isolates were substantially resistant, with MICs up to >256 mg/L. High MICs were not associated with particular ESBL types: 9/13 (69.2%) isolates with MICs >8 mg/L were positive for CTX-M group 1 enzymes, as were 85/140 (60.7%) of the strains with MICs ≤1 mg/L. It was concluded that possibly the more resistant isolates synthesized the larger amounts of ESBL, possessed different CTX-M variants or families, or presented multiple β-lactamases. This observation confirmed our previous findings, which showed that above 65% of *K. pneumoniae* isolates possessed two or more ESBL enzymes from different families. C/T was active against around 80% of *Enterobacteriaceae* isolates inferred to have reduced permeability without ESBL, AmpC, or carbapenemase activity—a group also widely susceptible (MIC  ≤ 1 mg/L, EUCAST) to cefotaxime (59.6%), ceftazidime (48.3%), and cefepime (56.2%) [17].

Sufficient activity of C/T against *K. pneumoniae* strains isolated from complicated intra-abdominal infections (87.3%) was also noted by Goodlet et al. [36], but it decreased to 63.6% in the ESBL producers group and 28.6% in MDR strains. Regarding the ESBL-producing strains, the mentioned study noted the variable activity of C/T that limits the use of this drug in an empirical therapy in patients with a high risk of infections caused by ESBL-positive strains.

In turn, Sutherland and Nicolau [26] noted 84.0% (58/69) of *K. pneumoniae* strains were sensitive to C/T. Higher activity was noted only to imipenem and meropenem, at 90.0%. The strains were derived from lower respiratory tract. On the other hand, Sader et al. [31], in a community-based study of isolates from Eastern and Western Europe conducted in 2012–2018, reported 47.3–66.7% (average 56.8%) and 82.2–89.6% (average 85.4%) of *K. pneumoniae* strains sensitive to C/T. The MIC_50_ and MIC_90_ values for strains derived from Eastern Europe were 1 mg/L and >32 mg/L, respectively, while the corresponding values for Western European strains reached 0.25 mg/L and 32 mg/L. Higher percentages of susceptibility were noted only for meropenem (78.6% and 90.3%), colistin (90.3%, 95.4%), and amikacin (75.9%, 90.9%). In the aforementioned study, the authors also showed data for susceptibility to C/T from Poland for Enterobacterales rods in general (32.7%, 621 strains), and it was the lowest percentage from Eastern European countries. The authors state that the results should be considered with great caution due to the different antibiotic policies and treatments for their use in different countries. The authors explained the significant differences between Eastern and Western Europe, a high percentage of ESBL-positive strains (EARS-Net data), and increasing resistance to antibiotics or the rapid clonal spread of resistance genes. Data from the EARS-Net [37] show that the percentage of resistant strains increase from the north of Europe to the south and from the west to the east of Europe. In turn, García-Betancur et al. [25], examining 610 *K. pneumoniae* strains isolated from patients from 5 Latin American countries, recorded 68.7% of strains sensitive to C/T, and 79.7% among 153 strains of ESBL-positive non-CRE were sensitive to C/T, which were the highest values among the β-lactam antibiotics, followed by imipenem—97.4%, meropenem—99.3%, and doripenem—100%.

On the other hand, Sader et al. [34] analyzed a big group of 3004 *K. pneumoniae* strains isolated from ICU and non-ICU patients. The authors detected 94.8% and 96.9% of strains susceptible to C/T, respectively, within these groups. For ESBL-positive Enterobacterales strains, the percentages were 86.6%, 94.1%, for MDR strains the percentages were 62.1% and 78.7%, while for XDR, the percentages were 14.3% and 17.0%, respectively. Higher percentages of isolates from both groups (ICU and non-ICU) were susceptible to ceftazidime/avibactam, meropenem/vaborbactam, amikacin (all above 99%), and to colistin (over 98%). In our study 18.6% of the analyzed *K. pneumoniae* strains isolated from ICU patients and 19.3% isolated from non-ICU patients were susceptible to C/T, and over 64% were detected as MDR and 13% as XDR, but the group of the analyzed strains was smaller (100 isolates).

In turn, Livermore et al. [17] showed that 97.6% out of 1296 *Klebsiella* spp. strains were susceptible to C/T, while for *K. pneumoniae*-producing ESBL, only 26.3% (255 strains) were susceptible. The corresponding values for *K. pneumoniae* AmpC hyper-producers were 51.0% (49 strains) and 12.5% (8 strains) for these producing ESBL and AmpC simultaneously. About half of the ESBL strains presented C/T MIC at the level of 2 mg/L, but for the others, it reached >256 mg/L. The authors stated that resistance does not have an association with specific ESBL enzyme. The authors proposed that resistant strains may produce large amounts of ESBL or other types of or different ESBL variants. In our study, all of the analyzed *K. pneumoniae* strains were ESBL-positive and only 30% of strains were susceptible to C/T.

In summary, ceftolozane–tazobactam may be an option in the therapy of infections caused by carbapenem-resistant *P. aeruginosa* strains, and this may also be an option for ICU patients. Moreover, as has been recently confirmed, EUCAST Rapid Antimicrobial Susceptibility Testing (RAST) is a reliable method to determine microbial susceptibility to ceftolozane–tazobactam for Gram-negative rods [38], which is of great importance for septic patients with infections of this etiology. Meanwhile, the infections caused by carbapenem-resistant *K. pneumoniae* strains should not be treated with C/T combination, mostly due to diversity, dissemination, and proficient synthesis of ESBLs, which may interfere with ceftolozane–tazobactam activity in vivo.

The data collected were obtained during routine diagnostic procedures applied in the Microbiology Department of University Hospital No. 1 in Bydgoszcz, Poland. Moreover, the institutional statement was waived due to the study design, which was conducted anonymously.

## 4. Materials and Methods

### 4.1. Bacterial Isolates and Their Origin

The overall number of 316 clinical strains was investigated, including 206 *P. aeruginosa* and 110 *K. pneumoniae* isolates. All the strains were isolated from clinical specimens derived from the patients of the University Hospital No. 1 in Bydgoszcz, Poland in 2018–2021. All of them were resistant or intermediate to at least one of the carbapenems (imipenem and/or meropenem for *P. aeruginosa* or additionally to ertapenem for *K. pneumoniae*) that present activity against both species and are used for the treatment of infections caused by them.

The detailed origin of the examined *P. aeruginosa* strains is presented in Appendix A, while the corresponding details for *K. pneumoniae* are presented in Appendix A.

### 4.2. Carbapenems Susceptibility Testing and Quality Control

During the antimicrobial susceptibility testing step, the following were used: the diffusion method on Mueller–Hinton Agar (Becton Dickinson, Frankfurt, Germany) and imipenem (10 µg) and meropenem (10 µg) discs plus ertapenem (10 µg) for *K. pneumoniae* only (Becton Dickinson, Germany). The results of the antimicrobial susceptibility tests were interpreted according to the European Committee on Antimicrobial Susceptibility Testing recommendations (EUCAST, Breakpoint tables for bacteria, Clinical breakpoints—bacteria v. 12.0) [22]. *P. aeruginosa* was purchased from American Type Culture Collection (ATCC 27853) and *E. coli* ATCC 25922 strains served as a susceptibility testing quality control.

### 4.3. Carbapenemase Activity Exclusion

The synthesis or absence of carbapenemases among the tested strains was previously excluded during standard diagnostic procedures and/or by using additional molecular studies. Initially, all of the examined strains were checked phenotypically for MBLs synthesis using the method described by Lee et al. [39] and by Yong et al. [40]. For the study purposes, the following methods were also used: Carba NP, CIM test, Phoenix NMIC-502 panels (Becton Dickinson), and molecular methods based on real-time polymerase chain reaction, and IVD tests to detect carbapenemase-encoding genes (VIM-/ IMP-, KPC-, or NDM-type; OXA-48 in CPE BD MAX Assay (Becton Dickinson); or KPC, VIM-type, NDM, OXA-48, and OXA-181 of eazyplex SuperBug CRE test (Amplex Diagnostics)). All the strains included in the study were carbapenemase-negative and characterized by the absence of carbapenemase synthesis potential by at least two methods (phenotypic and/or molecular) performed independently with compatible results obtained. Among the examined strains, all were resistant to at least one of the applied carbapenems or ticarcillin/clavulanate (according to carbapenemase-suspected strains) in the disc diffusion method, while all were simultaneously negative for the detection of an enzymatic resistance mechanism to carbapenems, regardless of the methodology applied.

### 4.4. Extended-Spectrum Beta-Lactamase Detection

An antimicrobial susceptibility testing and ESBL-type enzymes synthesis were determined using BD Phoenix™ M50 instrument (Becton-Dickinson, Franklin Lakes, NJ, USA) with NMIC-402 panels, both applied according to the manufacturer’s instructions.

### 4.5. Ceftolozane–Tazobactam Sensitivity Testing

Susceptibility of the strains to ceftolozane–tazobactam was evaluated quantitatively on Mueller–Hinton Agar (Becton Dickinson) with the application of paper strips with antimicrobial gradient (Liofilchem). The whole methodology was applied according to manufacturers’ instructions. The results of ceftolozane–tazobactam susceptibility tests were interpreted according to the European Committee on Antimicrobial Susceptibility Testing recommendations (EUCAST, Breakpoint tables for bacteria, Clinical breakpoints—bacteria v. 12.0) [22].

## 5. Conclusions

Based on the obtained results, ceftolozane–tazobactam in vitro activity is quite satisfactory against carbapenem-resistant *P. aeruginosa* but not against carbapenem-non-susceptible *K. pneumoniae* isolates. Therefore, other antimicrobials should be considered for the treatment of infections caused by carbapenem-resistant Enterobacterales rods, especially by *K. pneumoniae* isolates.

## Figures and Tables

**Figure 1 medicina-59-00518-f001:**
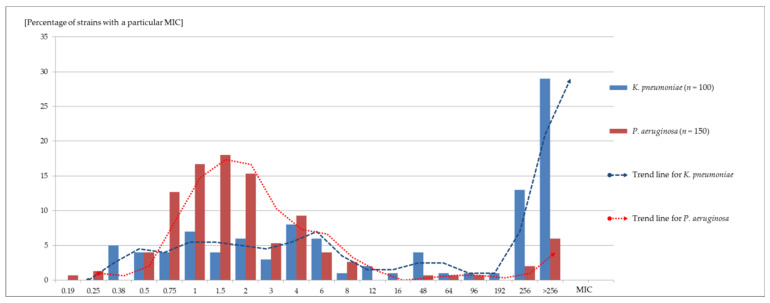
Ceftolozane–tazobactam MIC distribution in the groups of the examined isolates (*n* = 250).

## Data Availability

The data presented in this study are available on reasonable request from the corresponding author.

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
