# Peer review of "Satisfactory In Vitro Activity of Ceftolozane–Tazobactam against Carbapenem-Resistant Pseudomonas aeruginosa But Not against Klebsiella pneumoniae Isolates"

_medicina, 2023, doi:10.3390/medicina59030518_

Round 1
Author Response
We would like to thank the honorable Reviewer for carefully reading of the manuscript and providing all the valuable comments. All the suggestions found in the attached file have been taken into account and corrected in the current version of the manuscript. Please see below a detailed point-by-point response to all your comments.
Line 31-32 : Pseudomonas aeruginosa and Klebsiella pneumoniae isolates are currently one of the 31 most frequent causes of infections of immunocompromised patients ref
Line 32-33 : These rods are also among the most frequently isolated bacteria from infections caused by Gram-negatives, which include pathogens often causing nosocomial and opportunistic infections ref
Line 34-38 : P. aeruginosa shows intrinsic lack of susceptibility to a number of antimicrobials; both species easily acquire new resistance mechanisms ref.
These include resistance to carbapenems, often administered as the last choice drugs ref.
The increasing emergence of multi-drug resistant P. aeruginosa and K. pneumoniae strains limits the treatment options, which increases morbidity and mortality of the patients ref.
Line 42-46 : Ceftolozane-tazobactam (C/T) is a relatively new antimicrobial agent approved for the treatment of complicated urinary tract infections, including acute pyelonephritis ref.
It is also applied in the treatment of complicated intra-abdominal infections (in combination with metronidazole) and nosocomial pneumonia (including ventilator-associated pneumonia) ref.
The corresponding References have been added to the manuscript for all of the sentences above.
Line 284 : In the material and method, the ethics committee or the authorization of the Ministry of Health is missing
The relevant explanation has been added into Institutional Review Board Statement section of the manuscript.
Susceptibility of the strains to ceftolozane-tazobactam was evaluated quantitatively on Mueller-Hinton Agar (Becton Dickinson) with the application of paper strips with antimicrobial gradient (Liofilchem) : It would be better to refer to a recent study which used this same method to test the sensitivity of ceftolozane-tazobactam
To the best of our knowledge, Liofilchem is one of the two manufacturers and providers of ceftolozane-tazobactam strips for susceptibility evaluation quantitatively in Poland. The second one (bioMerieux) provides the strips but the only available publication (PMID: 29212704) concludes that, comparing these strips, categorical agreement for both test is very high (96.8%), no very major errors are observed between the tests, and no major errors are observed by them with the incidence of minor errors of 3.2%, giving altogether an excellent essential agreement. Therefore, we found this aspect irrelevant and decided to not include this information into the manuscript.
Line 286-287 : You should give more details of the origin of your bacteria samples
More details of the origin of bacterial strains are shown in the supplementary material (Tables S3 and S4). If the Reviewer still suggests moving them into the main body of the manuscript we are ready to do it. However, in our opinion they still fit better as Supplementary files.
Line 327-329 : Why didn't you use the double disc diffusion method on MH to test for the presence or absence of ESBL?
Although, according to manufacturer (Becton-Dickinson) Phoenix device and cards are validated for ESBLs evaluation and there is no need to check it, we did perform the testing for ESBLs presence for a selected group of strains using double disc diffusion method on MH agar to confirm the concordance of the results. The results were consistent in all of the studied strains, so we gave up with confirming these obvious results at some research point and decided not to include this section into the manuscript.
Reviewer 2 Report
I consider that the theme of the work” Satisfactory in vitro activity of ceftolozane-tazobactam against 2 carbapenem-resistant Pseudomonas aeruginosa but not against 3 Klebsiella pneumoniae isolates 4” is very relevant.
In these times, high resistance to antibiotics has become a very relevant issue due to its potential danger due to the presence of super resistant species.
However, I consider that this work is the beginning of an investigation that should be deepened further and analyzed with different methodologies to enrich the work presented.
Author Response
We would like to thank the honorable Reviewer for carefully reading of the manuscript and providing all the valuable comments. All the suggestions found in the attached file have been taken into account and corrected in the current version of the manuscript. Please see below a detailed point-by-point response to your comments.
I consider that the theme of the work” Satisfactory in vitro activity of ceftolozane-tazobactam against 2 carbapenem-resistant Pseudomonas aeruginosa but not against 3 Klebsiella pneumoniae isolates 4” is very relevant.
In these times, high resistance to antibiotics has become a very relevant issue due to its potential danger due to the presence of super resistant species.
We would like to thank the Reviewer for this estimating remark.
However, I consider that this work is the beginning of an investigation that should be deepened further and analyzed with different methodologies to enrich the work presented.
Indeed, further studies are necessary and planned to compare the results of C/T activity observed in this study with other new combinations of antibiotics/beta-lactamases inhibitors as well as other new antimicrobials, also non-beta-lactams. Moreover, in the future we also plan to investigate the susceptibility to C/T of other bacterial species.
Reviewer 3 Report
General comments
The manuscript by Sękowska et al. describes a study on the in vitro susceptibility of carbapenem-resistant P. aeruginosa and K. Pneumoniae strains, against ceftolozame-tazobactam combination. The authors also provide a detailed literature review in the discussion part, for P. aeruginosa, Enterobacteriaceae and K. pneumoniae isolates. The manuscript is in large parts well written; however, some parts of the discussion require careful examination and correction of the percentages and numbers mentioned. In this reviewer's opinion, the presented manuscript requires some modifications before a possible publication in Medicina.
Please find more detailed remarks below.
Specific comments
Abstract
Lines 19,20 : The sentence “… is also completed as the elements of brief review of the relevant literature” needs rephrasing.
Introduction
The introduction is well-written and provides an overview of the topic background.
Line 33: Please replace “Gram-negatives” with “Gram-negative bacteria”.
Line 48: Please mention each substance separately, for instance: “Ceftolozane-tazobactam consists of a new cephalosporin (ceftolozane) and a class A β-lactamase inhibitor (tazobactam).”
Line 53: Remove “also”.
Results
Lines 69-70: The authors mention a total of 150 P. aeruginosa strains in the results section, whereas in the abstract 206 P. aeruginosa strains are mentioned. Similarly, for K. pneumoniae they mention 110 strains in the abstract and 100 strains in the results sections, respectively. The reasons for this discrepancy are explained only in the M&M section, and thus, lines 293-298 should be moved to the results section for clarity purposes.
Discussion
The discussion part is well-written and offers a detailed overview of the literature, however it requires several corrections.
Overall: Mentioned concentrations should be either in mg/L, or μg/ml throughout the text.
Lines 113-144: Percentages should be mentioned instead of numbers.
Lines 120-121: The statement “...60.0% isolates within this group were resistant to β-lactams“ is incorrect. Sixty percent of the 10 BLR strains were susceptible to C/T (which corresponds to 6 strains). However, the isolates within the P. aeruginosa group that were resistant to β-lactams were 10 out of 45, thus 22.2%. The BLR, C/T susceptible strains were 6/45 and thus, only 13%.
Lines 138-139: Please rephrase to “Also, high percentages of susceptibility to colistin (above 99%) and to tobramycin (70.9-88.6%) were detected.”
Lines 141-142: Israel exhibited a higher percentage than Slovenia (98.2% versus 91.3%, respectively). Please correct accordingly.
Lines 143-144: Pfaller et al. mention 440 P. aeruginosa isolates, as well as that “Ceftolozane–tazobactam was the most potent (MIC50/90, 0.5/2 μg/mL) β-lactam agent tested against P. aeruginosa isolates, inhibiting 95.7% at an MIC of ≤4 μg/mL.” Please correct accordingly or provide another reference.
Lines 145-147: In the Pfaller study, the data mentioned for meropenem, cefepime, ceftazidime, and piperacillin/tazobactam, do not correspond to those mentioned in the text. The MIC50 values were 0.25, 2, 2, and 4 μg/ml, respectively. Therefore, based on the MIC50 value of C/T (0.5 μg/ml), it was 2-fold less active than meropenem, 4-fold more active than cefepime and ceftazidime and 8-fold more active than piperacillin/tazobactam. Moreover, Pfaller et al. mention that “Based on the MIC90 value, the potency of ceftolozane–tazobactam was equal to that of meropenem (MIC90, 2 mg/mL), 4-fold more active than cefepime (MIC90, 8 mg/mL), 8-fold more active than ceftazidime (MIC90, 16 mg/mL) and 32-fold more active than piperacillin–tazobactam (MIC90, 64 mg/mL)”. Could the authors please provide the numbers they used for their calculations?
Lines 147-150: Again, the percentages mentioned here, are not in line with those mentioned in the study by Pfaller et al.
Line 164, 166: The abbreviations “MBL”, “VEB” and “GES” should be defined.
Line 180: Please correct to “has been shown”.
Line 194: Please correct to “initial studies”.
Line 196: In the cited study, the mentioned Enterobacteriaceae isolates are 1019 and not 1878. All the percentages mentioned in this paragraph (Lines 194-207), also differ. Please check and clarify.
Line 217: Please correct “as” to “as were”.
Line 221: Please provide reference for the previous data.
Line 229: Please correct “author” to “study”. Also check spelling in the same sentence.
Line 232: Sutherland and Nicolau.
Line 252: The percentage mentioned refers to E. coli ESBL non-CRE. The correct percentage is 79.7% (153 strains).
Lines 253-254: These percentages are for P. aeruginosa strains, not K. pneumoniae. Please correct accordingly.
Line 257: These are ESBL-positive Enterobacterales, not specifically K. pneumoniae. Please clarify in the text.
Lines 259-261: Here, the authors refer only to K. pneumoniae strains again. Also the word “susceptible to” must be added. Please correct accordingly.
Line 265: The study by Livermore et al. is [13], not [5].
Line 266: 255 total strains (26.3%).
Lines 267-268: The correct numbers are 49 strains and 8 strains total, respectively. The time period should also be mentioned (2015-2016).
Line 271: Please remove “a thesis”.
Materials and Methods
The materials and methods section is well written.
References
Species names should be in italic font, for example please correct “Pseudomonas Aeruginosa” to “Pseudomonas aeruginosa”.
Author Response
We would like to thank the honorable Reviewer for carefully reading of the manuscript and providing all the valuable comments. All the suggestions found in the attached file have been taken into account and corrected in the current version of the manuscript. Please see below a detailed point-by-point response to all your comments.
Abstract
Lines 19,20 : The sentence “… is also completed as the elements of brief review of the relevant literature” needs rephrasing.
It has been rephrased.
Introduction
The introduction is well-written and provides an overview of the topic background.
Line 33: Please replace “Gram-negatives” with “Gram-negative bacteria”.
Line 48: Please mention each substance separately, for instance: “Ceftolozane-tazobactam consists of a new cephalosporin (ceftolozane) and a class A β-lactamase inhibitor (tazobactam).”
Line 53: Remove “also”.
All of the above sentences have been corrected in the manuscript.
Results
Lines 69-70: The authors mention a total of 150 P. aeruginosa strains in the results section, whereas in the abstract 206 P. aeruginosa strains are mentioned. Similarly, for K. pneumoniae they mention 110 strains in the abstract and 100 strains in the results sections, respectively. The reasons for this discrepancy are explained only in the M&M section, and thus, lines 293-298 should be moved to the results section for clarity purposes.
The mentioned part has been moved to Results section.
Discussion
The discussion part is well-written and offers a detailed overview of the literature, however it requires several corrections.
Overall: Mentioned concentrations should be either in mg/L, or μg/ml throughout the text.
Lines 113-144: Percentages should be mentioned instead of numbers.
Lines 120-121: The statement “...60.0% isolates within this group were resistant to β-lactams“ is incorrect. Sixty percent of the 10 BLR strains were susceptible to C/T (which corresponds to 6 strains). However, the isolates within the P. aeruginosa group that were resistant to β-lactams were 10 out of 45, thus 22.2%. The BLR, C/T susceptible strains were 6/45 and thus, only 13%.
Lines 138-139: Please rephrase to “Also, high percentages of susceptibility to colistin (above 99%) and to tobramycin (70.9-88.6%) were detected.”
All of the above have been changed in the manuscript.
Lines 141-142: Israel exhibited a higher percentage than Slovenia (98.2% versus 91.3%, respectively). Please correct accordingly.
It has been completed.
Lines 143-144: Pfaller et al. mention 440 P. aeruginosa isolates, as well as that “Ceftolozane–tazobactam was the most potent (MIC50/90, 0.5/2 μg/mL) β-lactam agent tested against P. aeruginosa isolates, inhibiting 95.7% at an MIC of ≤4 μg/mL.” Please correct accordingly or provide another reference.
Another reference has been added.
Lines 145-147: In the Pfaller study, the data mentioned for meropenem, cefepime, ceftazidime, and piperacillin/tazobactam, do not correspond to those mentioned in the text. The MIC50 values were 0.25, 2, 2, and 4 μg/ml, respectively. Therefore, based on the MIC50 value of C/T (0.5 μg/ml), it was 2-fold less active than meropenem, 4-fold more active than cefepime and ceftazidime and 8-fold more active than piperacillin/tazobactam. Moreover, Pfaller et al. mention that “Based on the MIC90 value, the potency of ceftolozane–tazobactam was equal to that of meropenem (MIC90, 2 mg/mL), 4-fold more active than cefepime (MIC90, 8 mg/mL), 8-fold more active than ceftazidime (MIC90, 16 mg/mL) and 32-fold more active than piperacillin–tazobactam (MIC90, 64 mg/mL)”. Could the authors please provide the numbers they used for their calculations?
Lines 147-150: Again, the percentages mentioned here, are not in line with those mentioned in the study by Pfaller et al.
The correct Reference has been added in the current version of the manuscript and we hope it is satisfactory now. Moreover, we found one of the sentences irrelevant and deleted it from the manuscript.
Line 164, 166: The abbreviations “MBL”, “VEB” and “GES” should be defined.
Line 180: Please correct to “has been shown”.
Line 194: Please correct to “initial studies”.
Line 196: In the cited study, the mentioned Enterobacteriaceae isolates are 1019 and not 1878. All the percentages mentioned in this paragraph (Lines 194-207), also differ. Please check and clarify.
Line 217: Please correct “as” to “as were”.
Line 221: Please provide reference for the previous data.
Line 229: Please correct “author” to “study”. Also check spelling in the same sentence.
Line 232: Sutherland and Nicolau.
All of the above have been corrected in the manuscript.
Line 252: The percentage mentioned refers to E. coli ESBL non-CRE. The correct percentage is 79.7% (153 strains).
Lines 253-254: These percentages are for P. aeruginosa strains, not K. pneumoniae. Please correct accordingly.
Line 257: These are ESBL-positive Enterobacterales, not specifically K. pneumoniae. Please clarify in the text.
Lines 259-261: Here, the authors refer only to K. pneumoniae strains again. Also the word “susceptible to” must be added. Please correct accordingly.
Line 265: The study by Livermore et al. is [13], not [5].
Line 266: 255 total strains (26.3%).
Lines 267-268: The correct numbers are 49 strains and 8 strains total, respectively. The time period should also be mentioned (2015-2016).
Line 271: Please remove “a thesis”.
All of the above have been corrected in the manuscript.
Materials and Methods
The materials and methods section is well written.
References
Species names should be in italic font, for example please correct “Pseudomonas Aeruginosa” to “Pseudomonas aeruginosa”.
All of the above have been corrected in the manuscript.
Round 2
Reviewer 2 Report
thank for the response